# Sonic Hedgehog-Gli1 Signaling and Cellular Retinoic Acid Binding Protein 1 Gene Regulation in Motor Neuron Differentiation and Diseases

**DOI:** 10.3390/ijms21114125

**Published:** 2020-06-09

**Authors:** Yu-Lung Lin, Yi-Wei Lin, Jennifer Nhieu, Xiaoyin Zhang, Li-Na Wei

**Affiliations:** Department of Pharmacology, University of Minnesota, Minneapolis, MN 55455, USA; yllin@umn.edu (Y.-L.L.); linxx637@umn.edu (Y.-W.L.); nhieu001@umn.edu (J.N.); zhan6240@umn.edu (X.Z.)

**Keywords:** ALS, CRABP1, chromatin remodeling, Gli, motor neuron, retinoic acid, sonic hedgehog, SMA

## Abstract

Cellular retinoic acid-binding protein 1 (CRABP1) is highly expressed in motor neurons. Degenerated motor neuron-like MN1 cells are engineered by introducing SOD^G93A^ or AR-65Q to model degenerated amyotrophic lateral sclerosis (ALS) or spinal bulbar muscular atrophy neurons. Retinoic acid (RA)/sonic hedgehog (Shh)-induced embryonic stem cells differentiation into motor neurons are employed to study up-regulation of *Crabp1* by Shh. In SOD^G93A^ or AR-65Q MN1 neurons, CRABP1 level is reduced, revealing a correlation of motor neuron degeneration with *Crabp1* down-regulation. Up-regulation of *Crabp1* by Shh is mediated by glioma-associated oncogene homolog 1 (Gli1) that binds the Gli target sequence in *Crabp1′s* neuron-specific regulatory region upstream of minimal promoter. Gli1 binding triggers chromatin juxtaposition with minimal promoter, activating transcription. Motor neuron differentiation and *Crabp1* up-regulation are both inhibited by blunting Shh with Gli inhibitor GANT61. Expression data mining of ALS and spinal muscular atrophy (SMA) motor neurons shows reduced CRABP1, coincided with reduction in Shh-Gli1 signaling components. This study reports motor neuron degeneration correlated with down-regulation in *Crabp1* and Shh-Gli signaling. Shh-Gli up-regulation of *Crabp1* involves specific chromatin remodeling. The physiological and pathological implication of this regulatory pathway in motor neuron degeneration is supported by gene expression data of ALS and SMA patients.

## 1. Introduction

Cellular retinoic acid-binding protein 1 (CRABP1) is a highly conserved cytosolic protein for binding retinoic acid (RA) with a high affinity [1]. The canonical function of CRABP1 is believed to bind RA thereby regulating its bioavailability and metabolism [2,3]. RA is an essential nutrient in adults and an endocrine factor/morphogen critical to central nervous system (CNS) development including pattern formation and neuron differentiation. In adult animals, CRABP1 mRNA is detected in certain tissues/organs such as skin, heart, liver, adipose tissues, and the brain. In developmental stages, it is most highly detected in the developing CNS, especially at the stages of E9.5–E12.5 in mice [4]. The nervous system specificity of *Crabp1* gene during the developmental stages is of most interest. Molecular studies have revealed multiple regulatory regions within a 3 kb sequence upstream of the transcription initiation site (TIS) of the mouse *Crabp1* gene [5,6].

Within this 3 kb upstream region, there exist the minimal promoter containing five Sp1 binding sites (GGGCGG boxes), and several conserved regulatory sequences such as an AP1 site, nine pairs of inverted repeats, and one hormone response element (HRE) that mediates either thyroid hormone-induced activation or RA-triggered suppression of this gene [5,6,7]. The GC-rich region is subjected to cell context-dependent DNA methylation, which contributes to its epigenetic silencing [8]. The HRE is responsible for its bi-directional regulation by thyroid hormones and RA, which contributes to specific chromatin remodeling of this promoter facilitated by a mediator-containing chromatin remodeling machinery and coactivator PCAF or corepressor RIP140 [9]. In searching for the brain/neuron specific activity of this gene, we have employed transgenic mice as the reporter system [1,10,11,12], and identified a brain/neuronal specific regulatory promoter within approximately 500 base pairs (bps) upstream of TIS. This region contains only the minimal promoter (Sp1 sites) and an approximately 200 bps upstream sequence [5]. In spite of extensive studies of *Crabp1* gene, the mechanism mediating its neuron specific expression has remained a mystery. This current study aims to identify and determine the mechanism, as well as the signaling pathway, underlying the regulation of *Crabp1* gene’s motor neuron specificity, and to address whether this regulation is associated with human diseases (see below).

To this end, we previously documented that *Crabp1* knockout (CKO) adult mice exhibited multiple phenotypes, such as augmented hippocampal learning ability, increased adipose tissue hypertrophy, and deteriorated cardio-pathology in an isoproterenol-induced heart failure model [13,14,15]. These are consistent with the scope of its expression in adult stages. As human gene expression data have become increasingly available, it is interesting to recognize that CRABP1 expression is down-regulated in motor neurons of proximal spinal muscular atrophy (SMA) cells and animal models [16]. Clinical gene expression data have revealed that CRABP1 expression is also down-regulated in the spinal motor neurons of sporadic amyotrophic lateral sclerosis (SALS) patients [17]. A healthy state in SH-SY5Y cells, rescued with an ALS candidate peptide drug GM604, correlates with up-regulation of *Crabp1* gene expression [18]. These observations all suggest a correlation of *Crabp1* gene dys-regulation (especially down-regulation) with motor neuron disorders such as SMA and ALS. This further prompted us to carry out the current study to determine how *Crabp1* gene is specifically up-regulated in motor neurons and whether dysregulation in this gene is associated with diseases.

As introduced above, approximately 500 bps upstream of the TIS of the mouse *Crabp1* gene is sufficient to drive brain/neuron specific expression of a lacZ reporter mimicking endogenous *Crabp1* gene expression pattern in transgenic mice [19]. This 500 bps sequence contains approximately 200 bps of uncharacterized sequences and a minimal promoter (300 bps). One prominent feature of this 200 bps upstream sequence is a potential binding site for the transcription factor glioma-associated oncogene homologs (Gli1, 2, and 3). Glis are known to mediate the action of sonic hedgehog (Shh), a secreted signaling peptide critical for embryonic pattern formation and development, especially for the brain and spinal cord. Shh binds to the transmembrane receptor, protein patched homolog 1 (Ptch1), which weakens the inhibition of smoothened homolog (SMO) and then activates Glis [20]. Shh and Shh signaling agonists are also widely used to induce embryonic stem cells (ESCs) and human induced pluripotent stem cells (iPSCs) differentiation into motor neurons. Given the presence of a conserved Gli binding site in the 200 bps upstream region, and the effects of Shh/Gli signaling in neurons, this current study focuses on the Shh/Gli pathway to examine how *Crabp1* gene is up-regulated in motor neurons and determines whether dysregulation in *Crabp1* and Shh/Gli signaling is associated with human diseases of motor neurons.

## 2. Results

### 2.1. Crabp1 is Highly Expressed in Spinal Motor Neurons

We decided to first examine the CNS in adult mice to monitor *Crabp1* gene expression. We dissected brain regions and the spinal cord for Western blot analyses of CRABP1 expression as shown in Figure 1a. Clearly, the CRABP1 level is highest in the spinal cord, followed by the medulla and pons in the brain. Therefore, we focused on the spinal cord for more careful examination of cell type specificity of CRABP1 expression by immunostaining. We first examined the lumbar spinal cord for the expression of a pan neuronal marker NeuN and CRABP1 (Appendix A). The result showed that all CRABP1-exressing cells in the spinal section are neurons (NeuN-positive); however, not all NeuN-positive cells are CRABP1-positive (white arrow), indicating that CRABP1 is expressed in the neuronal population of the spinal cord. Further staining with motor neuron marker ChAT revealed that CRABP1 was most abundantly expressed in motor neurons. As shown in Figure 1b, under immunohistochemical staining, CRABP1 protein signals specifically overlap with signals of ChAT, but not of astrocyte marker GFAP. Quantification of these sections showed that greater than 95% of ChAT-positive cells are CRABP1-positive (Appendix A). These results clearly reveal that *Crabp1* gene expression is mostly active in spinal motor neurons. This is consistent with the gene-disease association data that suggest association of altered CRABP1 expression with diseases in motor neurons (see later).

### 2.2. Crabp1 Expression is Down-Regulated in Diseased Motor Neurons

As introduced earlier, recent expression data have shown that CRABP1 level is lower in conditions of motor neuron degeneration such as ALS and SMA. We thus exploited a motor neuron-like cell line MN1 to examine the relationship of a reduction in CRABP1 level and neuron degeneration. G93A is one of the most commonly identified superoxide dismutase 1 (SOD1) mutations in familial ALS [21]. We employed a strategy to engineer healthy vs. diseased (ALS) motor neuron models by expressing the wild type SOD1 (SOD1^WT^) or mutated SOD1 (SOD^G93A^). As shown in Figure 2a, diseased MN1 cells (containing the mutant, SOD^G93A^) have a significantly lower level of CRABP1, as compared to the healthy counterpart, SOD1^wt.^. We further exploited another motor neuron disease model, spinal and bulbar muscular atrophy (SBMA), which is a currently untreatable motor neuron disease caused by the expansion of a polyglutamine (polyQ) repeat in the androgen receptor (AR). We employed the same strategy to engineer healthy vs. degenerated MN1 cells to model SBMA neurons [22]. MN1 cells containing AR-24Q (control) remain healthy, whereas MN1 cells containing the diseased version, AR-65Q, become degenerated in cultures. As shown in Figure 2b, the diseased SBMA/MN1 neurons, AR-65Q, also have a dramatically reduced CRABP1 level, as compared to the healthy control, AR-24Q. These results, in two motor neuron disease models, consistently show that *Crabp1* gene activity is positively correlated with a healthy state in motor neurons, whereas down regulation of the *Crabp1* gene is correlated with degeneration in motor neurons, such as those in ALS and SBMA.

### 2.3. Sonic Hedgehog Signaling Up-Regulates Crabp1 in Motor Neuron Differentiation

To understand how the *Crabp1* gene is highly and specifically activated in motor neurons, we employed an ESC-motor neuron differentiation model. This model utilizes a specific cocktail to induce ESC differentiation into motor neurons. The most common cocktail contains two key factors, RA and Shh, added after embryo body (EB) formation. Figure 3a shows the standard motor neuron differentiation procedure that begins with ESC, followed by EB formation, differentiation cocktail, neurosphere formation, and motor neuron differentiation. In order to compare and determine the effects of RA and Shh on motor neuron differentiation and *Crabp1* expression, we included four experimental groups (CON, RA, Shh, and RA+Shh). In this ESC differentiation system, RA induces EBs differentiation into different types of neurons [23,24]. Shh further facilitates RA-exposed EBs differentiation into motor neurons [25]. Without first forming EBs, RA induces ESC differentiation into mixed populations of cells. Figure 3b validates the efficiency of motor neuron differentiation, identified with two markers Hb9 and ChAT. As predicted, RA+Shh is indeed most effective in inducing motor neuron differentiation. We then monitored the expression of CRABP1 (Figure 3c). It is interesting that RA+Shh treatment indeed increases CRABP1 level, but Shh alone most robustly activates the *Crabp1* gene. This is consistent with our previous finding that RA is generally a suppressive agent for *Crabp1* gene, acting through the bi-directional HRE in the 3 kb upstream region [9]. This result also suggests a tight control for CRABP1 level needed for motor neurons, ensuring the maintenance of an optimal amount of CRABP1 that is most desirable for healthy motor neurons. Nevertheless, we conclude that in motor neurons Shh is the key factor for up-regulating *Crabp1* gene.

To validate this notion, we employed a selective inhibitor of Gli1/2, GANT61. Gli transcription factors are the direct signal mediators of Shh to activate target genes. As shown in Figure 3d, GANT61 clearly suppresses motor neuron differentiation from ESC, supported by the reduction in motor neuron markers Hb9 and ChAT. Therefore, blunting Shh signaling with GANT61 reduces the CRABP1 level, supporting the specific effect of Shh-Gli signaling in up-regulating the *Crabp1* gene.

### 2.4. Gli1 Directly Binds to Its Chromatin Target on Crabp1, Inducing Juxtaposition with the Minimal Promoter to Up-regulate Crabp1 Expression

To determine the mechanism of how Shh-Gli signaling activates *Crabp1* gene transcription, we adopted one of our established procedures monitoring chromatin conformational changes [9]. This procedure is based upon a transcription factor binding assay, chromatin immunoprecipitation (ChIP). The experiment was designed to address three key issues. First, Gli, activated by Shh, may directly bind to the predicted chromatin target in the 200 bps region. Second, it is known that Gli can be associated with Sp1 [26]; therefore Gli binding may induce chromatin conformational change by triggering chromatin juxtaposition with the minimal promoter that has Sp1 sites. If this were to happen, Gli would also be physically associated with the minimal promoter region even if this region does not contain its target sequence. Thirdly, it is important to determine whether this happens only in differentiated motor neurons (with differentiation cocktail containing RA+Shh). These issues are addressed in ChIP assays that monitor chromatin regions where Gli can be physically associated.

Figure 4a shows the map of the 500 bps region that encodes the neuron-specific regulatory activity. This region contains a Gli binding site, immediately upstream of the minimal promoter that contains only Sp1 sites. In the ChIP assays, binding to the Gli site is indicated by the detection of a 160 bps PCR fragment (filled box) that contains the Gli site, whereas binding to the minimal promoter is indicated by the detection of a 189 bps PCR fragment (striped box) that contains only Sp1 sites.

Gli1 is known to be increasingly expressed in developing lumbosacral spinal cord [27]; therefore a specific Gli1 antibody was used in these ChIP experiments. ChIP data are shown in Figure 4b (Gli binding site) and 4c (minimal promoter). Clearly, Gli1 binding to its target site is robust in EBs-differentiated motor neurons (EB D4, RA+Shh treated) but not detected in EBs differentiated cells treated with RA alone (EB D4, RA) (4b right panel). Importantly, in control group (ESC), Gli1 binding was not detected in either RA or RA+Shh treated cells (4b left panel) (note the difference in the Y scale bars). This result shows that Gli1 indeed increasingly binds to its chromatin target only in differentiating motor neurons (EB D4, RA+Shh treated), but not in RA alone-differentiated cells (RA treated) or undifferentiated (ESC) cells.

The result of Gli1 association with the minimal promoter that contains only Sp1 sites is shown in Figure 4c. Again, Gli1 is also increasingly associated with the *Crabp1* minimal promoter in differentiating motor neurons (EB D4, RA+Shh treated, right panel) but not in cells differentiated with RA alone (EB D4, RA treated, right panel) or undifferentiated (ESC, left panel) cells. This result shows that Gli1, activated by Shh to bind the Gli site-containing chromatin, indeed can also be physically associated with the minimal promoter, supporting chromatin juxtaposition of the Gli1 target site (the 200 bps upstream fragment) with the minimal promoter (Sp1 sites).

Figure 4D depicts the model of chromatin remodeling occurred on *Crabp1* promoter region, triggered by Gli1 binding to chromatin after Shh stimulation, which leads to *Crabp1* gene activation in differentiating motor neurons. This model explains how Shh-Gli1 induces chromatin juxtaposition on the neuron-specific promoter region of *Crabp1* gene, allowing transcription factors to activate the preinitiation complex (PIC) for active transcription in differentiating neurons.

### 2.5. Dysregulation of CRABP1 and Shh Signaling Components in Human Motor Neuron Diseases

Given the clear effects of Shh-Gli1 signaling on *Crabp1* gene activation in differentiating motor neurons as described above, it is of interest to examine and validate whether this motor neuron-specific regulatory event is defective in diseased motor neurons. Literature has most extensively reported two human motor neuron diseases, ALS and spinal muscular atrophy (SMA). We carefully carried out data mining of publicly available human expression data sets of ALS and SMA. Very interestingly, in both ALS and SMA patients, CRABP1 expression is significantly reduced (dropped to 0.12 and 0.01 fold, respectively, as compared to healthy levels) (Figure 5a). In both ALS and SMA studies, reduced *Crabp1* gene expression was validated with quantitative RT-PCR. [17,28]. This correlation strongly supports our proposition that *Crabp1* up-regulation in motor neurons is physiologically relevant, and that dysregulation (specifically down-regulation) of this gene is strongly associated with motor neuron disease conditions.

Since Shh-Gli1 signaling plays an intimate role in up-regulating *Crabp1* and facilitating motor neuron differentiation, we further examined the expression data of Shh-Gli signaling components that are available in the RNA-seq data from the SMA study (GEO Accession: GSE108094) by Rizzo et al., 2019. It appears that the mRNA levels of Shh and its signaling components including LRP2 and SMO are significantly reduced in diseased neurons. While the reduction in certain Shh signaling components is not statistically significant, they all exhibit the same trend of decreased expression in SMA. Interestingly, the level of Gas1, one component generally reduced by Shh activation [29], is correspondingly increased. These human data further support the notion that dysregulation in *Crabp1* and Shh signaling is associated with a diseased condition in motor neurons.

Figure 5c depicts the model of Shh signaling pathway regulating *Crabp1* gene expression. Dysregulated Shh signaling components associated with diseases, as revealed from the patients’ data shown in panel B, are marked with solid (increase, for Gas1) or empty (decrease, for Shh, LRP2, SMO, PTCH1, SUFU, and Glis) circles.

## 3. Discussion

The present study provides the evidence for the association of Shh-Gli1 regulation of *Crabp1* expression with motor neuron differentiation, and that down-regulation of *Crabp1*, as well as the Shh-Gli signaling, is associated with motor neuron diseases in humans, such as ALS, SBMA, and SMA [17,28]. The study further determines the underlying mechanism of up-regulating *Crabp1*, which involves chromatin conformational changes in the neuron-specific promoter of *Crabp1* gene by Shh-activated Gli1.

As clearly shown, in the CNS CRABP1 is most highly expressed in spinal motor neurons. The correlation of dysregulation in this gene with motor neuron diseases suggests certain physiological activity of CRABP1 in motor neurons. The canonical activity of CRABP1 is believed to bind RA thereby regulating (reducing) RA’s bioavailability and facilitating RA metabolism [2,3]. The rising CRABP1 level in differentiating neurons would predict a reduction in RA concentrations as neurons are increasingly differentiated in the system. However, this presents a paradox, because RA is generally required for neuronal differentiation. Therefore, whether CRABP1 indeed acts via the presumed activity to reduce RA concentration in motor neuron differentiation process remains to be examined. To this end, our studies of CKO mice have provided some interesting insights. The initial CKO studies (which were generated in the sv129 background) reported no defects in the CRABP1-depleted mice [30,31]. However, our studies of CKO mice that were generated in the C57/BL6 background have revealed multiple phenotypes in adult mice, including increased hippocampal neurogenesis and improved learning ability [15], increased adipocyte hypertrophy [13], and cardio-pathology in isoproterenol-induced heart failure [14]. All of these phenotypes are attributable to certain newly identified, non-canonical activities of CRABP1, which include its abilities to modulate ERK or CAMKII signaling pathways [14,32,33]. Therefore, the exact physiological role for CRABP1 in motor neurons remains to be rigorously examined.

RA and Shh are two key factors for motor neuron differentiation from ESC. RA is known to play important roles in neuronal differentiation via its activities mediated by nuclear RA receptors [23,34,35]. Shh is known to contribute to the differentiation specification toward motor neurons [36]. Additionally, Shh is required in multiple stages of neural development including neural stem cell differentiation, neural progenitor cell specification, cell proliferation, synapse formation and axon guidance [37,38,39]. Shh activates its downstream signaling pathway, ultimately activating Gli transcription factors which can regulate a wide spectrum of target genes that play roles in numerous processes for cell growth, differentiation, and functions. In vivo, the production of motor neurons is affected by Shh at least at two critical stages: naïve neural plate cells converted into ventralized progenitor cells, and ventralized progenitor cells differentiated into motor neurons [40]. It remains to be determined whether Shh also similarly regulates *Crabp1* gene in these various neural developmental processes in vivo.

Shh signaling is initiated from transmembrane receptor PTCH1 ultimately activating transcription factors Glis to regulate target genes. The Gli family includes Gli1, Gli2, and Gli3; Gli1 and Gli2 primarily act as transcriptional activators and Gli3 mostly acts as a transcriptional repressor. In developing motor neurons, Gli1 seems to be the primary mediator for Shh since it is most highly expressed in this system. It is tempting to speculate that, at the molecular level, Shh signaling ultimately drives specific chromatin conformational change of neuron-specific promoters, such as that of *Crabp1*, by inducing chromatin juxtaposition of the neuron-specific regulatory region with the minimal promoter. It would be interesting to examine whether this also occurs in the activation of other neuron-specific genes that contain Gli-binding regulatory elements. On the contrary, RA, a well-documented transcription factor that also can induce chromatin remodeling, fails to induce such a chromatin remodeling event or *Crabp1* gene activation, as shown in Figure 4. This further highlights the significance of Shh signaling in this neuron-specific chromatin remodeling process as exemplified by *Crabp1* gene activation. As to the functional role of Crabp1 in motor neurons, this would require much more comprehensive studies using rigorous genetic, neurological, and cell biological approaches in the future.

## 4. Materials and Methods

### 4.1. Animal Experiments

C56BL/6J mice from Jackson Laboratory were maintained in the animal facility of the University of Minnesota, in a temperature-controlled room (22 ± 1 °C) on a 14/10 light dark cycle (lights on/off at 0600/2000) with ad-lib food and water. Experimental procedures were conducted according to NIH guidelines and approved by the University of Minnesota Institutional Animal Care and Use Committee. All efforts were made to reduced animals’ suffering and the number of animals used.

### 4.2. Western Blotting

Eight week old mice were used in this experiment. The mice were sacrificed by CO_2_ and the different brain areas, including the cortex, hippocampus, thalamus, cerebellum, spinal cord, pons, and medulla, were isolated. Western blotting was conducted as described [41] using anti-β-actin (SC-47778, Santa Cruz, Dallas, TX, USA), anti-Crabp1 (C1608, Sigma-Aldrich, St. Louis, MO, USA), and anti-NeuN (MAB377, Millipore, St. Louis, MO, USA).

### 4.3. Immunohistochemistry

Eight week old mice were perfused with PBS containing 4% paraformaldehyde. Lumbar spines were removed, fixed in 4% paraformaldehyde for 2 h, and immersed in 30% sucrose for 24 h at 4 °C. Coronal sections were obtained in 20-μm-thick slices. PBS-washed slices were treated with a blocking solution containing 0.2% Triton X-100, 1% bovine serum albumin, and 5% goat serum in PBS for 60 min at room temperature, incubated with primary antibodies, including CRABP1 (C1608, Sigma–Aldrich; 1:400), ChAT (AB144P, Millipore-Sigma; 1:1000), and GFAP (AB5804; Millipore–Sigma, 1:1000) diluted in blocking solution at 4 °C overnight, and incubated with fluorochrome-conjugated secondary antibody and 4′,6-diamidino-2-phenylindole in the dark for 1 h. Fluorescent images were acquired under an Olympus FluoView 1000 IX2 upright confocal microscope.

### 4.4. Cell Culture and Disease Models

Murine motor neuron-neuroblastoma cell line (MN1) was kindly provided by Dr. Ahmet Hoke (The Johns Hopkins University, Baltimore, MD). MN1 cells were maintained with DMEM (11965, Gibco, Baltimore, MD, USA) supplemented with 10% fetal bovine serum (S11150, Atlanta, MN, USE), and 1% penicillin and streptomycin (25140, Gibco).

ALS cell disease model: Plasmid plv-AcGFP-SOD1 WT (27138, Addgene, Cambridge, MA, USA) and G93A (27142, Addgene) were purchased from Addgene. MN1 cells transfection was performed by Lipofectamine 3000 (L3000015, Fisher Scientific, Waltham, MA, USA) and conducted following the manufacturer’s instruction. Cells were harvested for analysis 2 days after transfection.

SBMA cell disease model: MN1 cells stably expressing the human androgen receptor protein with 24 polyglutamine residues (AR-24Q) or 65 polyglutamine residues (AR-65Q) were kindly provided by Dr. Kenneth H. Fischbeck (NIH, MD) [22]. Cells were harvested for analysis 2 days after subculture.

### 4.5. ESCs Culture and Differentiation into Motor Neurons

CJ7 mouse embryonic stem cells were used in this study. ESCs were grown on a primary mouse embryonic fibroblast feeder layer (Millipore) in 10 cm tissue culture dishes. ESCs were cultured and maintained with ESC medium (DMEM (11960, Gibco, MD, USA) supplemented with 15% StasisTM Stem Cell FBS (100-125, Gemini bio, West Sacramento, CA, USA), 1% ESGRO^®^ Recombinant Mouse LIF (ESG1107; Millipore Sigma–Aldrich), 1% L-glutamine (25030, Gibco), 1% nonessential amino acids (11140, Gibco), 5 µL/500 mL of β-mercaptoethanol (M3148, Sigma–Aldrich), and 0.2% penicillin/streptomycin (25140, Gibco)].

We followed the ESC-motor neuronal differentiation procedure from Gibco (Pub. No. MAN0016688) with modifications as outlined in Figure 3a. Briefly, ESCs were dissociated with 0.25% trypsin (25200, Gibco) and placed into a tissue culture dish coated with 0.1% gelatin (G1890, Sigma–Aldrich). After 30 min, the floating ESCs were transferred to a petri dish containing EB medium (45% DMEM/F-12 (10565, Gibco) and 45% neurobasal^TM^ medium (21103, Gibco) supplemented with 10% knockout^TM^ serum replacement−multi-species (A31815, Gibco), and 0.1% β-mercaptoethanol (21985, Gibco)). After two days, EBs were re-suspended in MN differentiation medium (EB medium supplemented with 0.5 µM RA (R2625, Sigma–Aldrich) and/or 200 ng/mL Shh (78066, Vancouver, Stemcell, Canada). After 2 days, the differentiated EBs were fully dissociated with Accumax^TM^ (A7089, Millipore). The single cells were suspended in MN differentiation medium and plated at the concentration of 4 × 10^6^/well in 6 well plate coated with 360 µg/mL Matrigel Basement Membrane (A1413301, Thermo Fisher Scientific). Half of the MN differentiation medium was changed every 2 day. GANT61 at a concentration of 10 µM (G9048, Sigma–Aldrich) was used to inhibit Gli activity.

### 4.6. Quantitative Real-time PCR (qPCR)

Total RNA was isolated using TRIzol (Invitrogen, Carlsbad, CA, USA), cDNA was synthesized using Omniscript RT kit (QIAGEN, Germantown, CA, USA), and qPCR was performed using SYBR-Green (Agilent, Santa Clara, CA, USA) and detected with Mx3005 P (Agilent). The primers were: ChAT, forward 5′-ACTGGGTGTCTGAGTACTGG-3′, reverse 5′-TTGGAAGCCATTTTGACTAT-3′; Hb9, forward 5′-TTCCAGAACCGCCGAATGAA-3′, reverse 5′-CCTTCTGCTTCTCCGCCTC-3′; Crabp1, forward 5′-ACCTGGAAGATGCGCAGCAGCGAG-3′, reverse 5′-TAAACTCCTGCATTTGCGTCCGTCC-3′.

### 4.7. Chromatin Immunoprecipitation (ChIP) Assay

The ChIP assay was conducted following the manufacturer’s instructions (#9002, Cell Signaling, Danvers, MA, USA). Briefly, cells were cross-linked with 1% formaldehyde and quenched by 125 mM glycine. Cells were lysed and chromatin were digested with a micrococcal nuclease. Digested chromatin were precipitated with Gli1 antibody (NB600-600, Novus, St. Charles, MO, USA) and protein A agarose beads. Precipitated chromatin was eluted, digested with proteinase K, and underwent reversal of the cross-link. DNA samples were then purified and analyzed by quantitative PCR. Primer sequences for the minimal promoter on CRABP1 were: forward 5′-CCAGGGGAGAGCAAGTTCC-3′, reverse 5′-CTTGAGTCGCTAGGGTAG-3′. Primer sequences for Gli1 binding sites on CRABP1 were: forward 5′-GTAGAGAAAGAATGTCGCG-3′, reverse 5′-GGAACTTGCTCTCCCCTGG-3′.

### 4.8. Data Mining of crabp1 and Shh in Human Motor Neuron Disease

A literature search identified two studies using diseased human spinal motor neurons (MNs) from sporadic amyotrophic lateral sclerosis patients (SALS) and MNs derived from induced pluripotent stem cells (iPSCs) of spinal muscular atrophy (SMA) patients [17,28]. In the SALS study, DNA microarray analysis was performed by Jiang et al. using GenePix Pro analysis software to identify differentially expressed genes in micro-dissected SALS motor neurons. For Figure 5a, Crabp1 was identified by GenePix Pro as the most significantly down-regulated gene in SALS motor neurons, reported as a 0.12-fold reduction. Fold change was calculated by taking the mean values from the motor neurons of 5 individual SALS patients divided by those of 5 healthy controls. In the SMA study, RNA-seq was performed by Rizzo et al. using CuffDiff2 analysis to identify differentially expressed genes in SMA iPSC-derived motor neurons [42]. For Figure 5a, Crabp1 SMA fold-change was calculated by dividing the raw fragments per kilobase of transcript per million (FKPM) values for SMA over the values for healthy motor neurons, reported as a 0.01-fold reduction. For Figure 5, raw FKPM values are reported. FKPM values were generated from the motor neurons derived from two individual SMA patients and two individual healthy controls. CuffDiff2 analysis identified that the differential expression of CRABP1, Shh, GAS1, LRP2, and SMO were significant in SMA. All raw FKPM values were provided by Rizzo et al. in their Appendix A and Gene Expression Omnibus (GEO) dataset (accession: GSE108094).

### 4.9. Statistical Analyses

Statistical differences between groups were determined by two-way analysis of variance (ANOVA) followed by Bonferroni’s post hoc test in each differentiation stage of Figure 3. Independent-sample *t*-tests were used to compare two independent groups in Figure 3 and Figure 4. Statistical analyses were performed by SPSS 17.0. All tests were performed at a significance level of *p* < 0.05, and data were presented as the mean ± SEM.

## Figures and Tables

**Figure 1 ijms-21-04125-f001:**
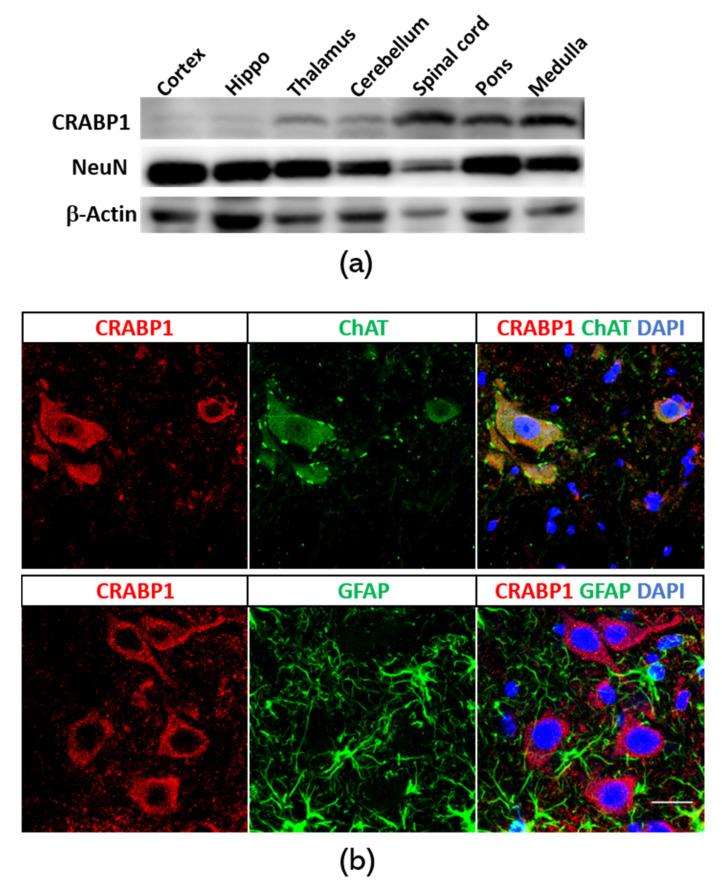
CRABP1 expression in the CNS. (**a**) Western blot analyses of CRABP1 in the CNS. (**b**) Confocal microscopy images showing signals of CRABP1 (red), spinal motor neuronal marker ChAT (green; upper middle), astrocyte marker (green; lower middle), and cell nuclei (DAPI). Scale bar = 20 µm.

**Figure 2 ijms-21-04125-f002:**
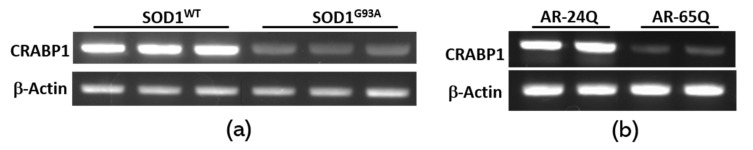
Crabp1 expression in motor neuron disease cell models. (**a**) CRABP1 mRNA levels in SOD1^WT^ and SOD1^G93A^ transfected MN1 cells (modeling ALS). (**b**) CRABP1 mRNA levels in AR-24Q and AR-65Q transfected MN1 cells (modeling SBMA). Abbreviations: ALS = amyotrophic lateral sclerosis, SBMA = spinal and bulbar muscular atrophy.

**Figure 3 ijms-21-04125-f003:**
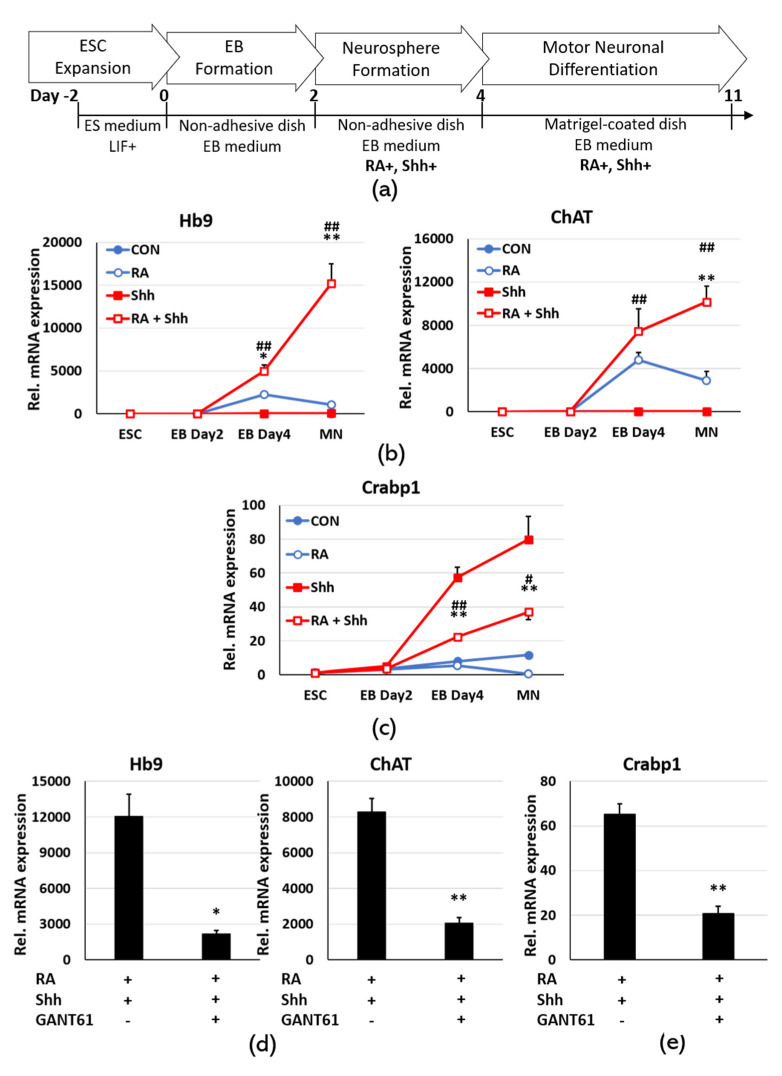
The expression of motor neuron markers and CRABP1 in embryonic stem cells (ESC) differentiation into motor neurons. (**a**) ESC-motor neuron differentiation procedure. (**b**) The results of qPCR analyses of motor neuron markers Hb9, ChAT. (**c**) The results of qPCR analyses of CRABP1, *n* = 3/group. The statistic results are presented as means ± SEM. * *p* < 0.05, ** *p* < 0.01; compared RA+Shh to RA only group. ^#^
*p* < 0.05, ^##^
*p* < 0.01; compared RA+Shh to Shh only group. (**d**) The results of qPCR analyses of Hb9, ChAT, and CRABP1 with and without the Gli1/2 inhibitor, GANT61 (*n* = 3/group). The results are presented as means ± SEM. * *p* < 0.05, ** *p* < 0.01. Abbreviations: EB = embryo body, RA = retinoic acid, Shh = sonic hedgehog.

**Figure 4 ijms-21-04125-f004:**
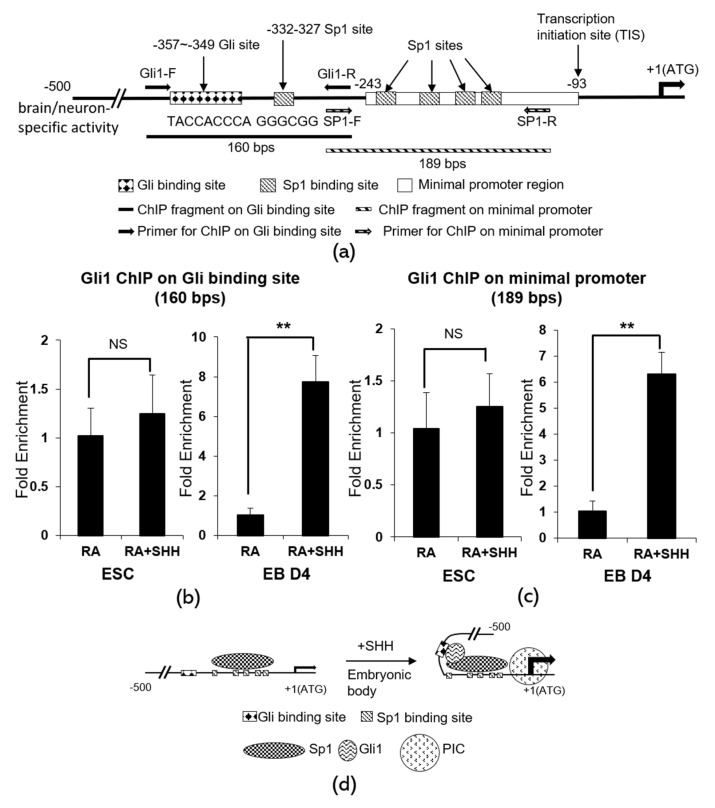
ChIP assays for Shh activated Gli-induced chromatin conformational change on the *Crabp1* promoter. (**a**) The map of neuron-specific promoter of *Crabp1* gene and the design of ChIP assays. Gli1-F, forward primer for the Gli site-containing fragment; Gli1-R, reverse primer for the Gli site-containing fragment; SP1-F, forward primer for the minimal promoter fragment; SP1-R, reverse primer for minimal promoter fragment. The Gli-binding chromatin is indicated by 160 bps PCR fragment. The minimal promoter containing Sp1 sites is indicated by 189 bps PCR fragment. (**b**) ChIP results of Gli1 physical association with the Gli binding site (160 bps fragment) in embryo body (EB D4, right) vs. embryonic stem cells (ESC, left). (**c**) ChIP results of Gli physical association with the minimal promoter (189 bps fragment) in embryo body (EB D4, right) vs. embryonic stem cells (ESC, left). Results are presented as means ± SD; * *p* < 0.05, ** *p* < 0.01; compared RA+Shh to RA only group; NS, not significant. (**d**) A molecular model depicting Shh activation of Gli1-induced chromatin juxtaposition of the Gli-binding region with the Sp1-binding minimal promoter of *Crabp1* gene. Change in this chromatin conformation brings the Gli1/Sp1 activator complex to the proximity of preinitiation complex (PIC), thereby facilitating transcription activation.

**Figure 5 ijms-21-04125-f005:**
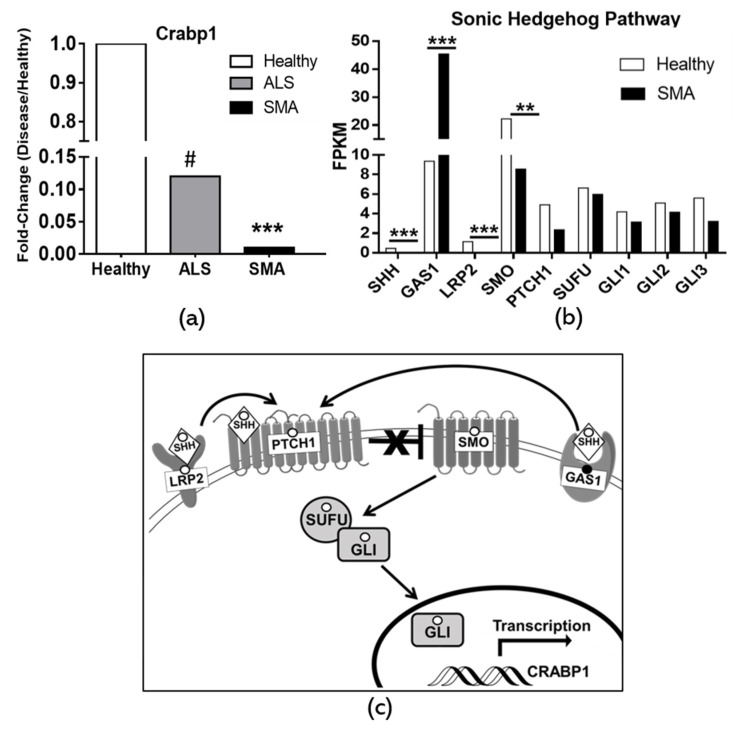
Dysregulation in CRABP1 and Shh signaling components in human motor neuron diseases. (**a**) Crabp1 gene expression is significantly reduced in patient-derived motor neurons. CRABP1 level is reduced to a 0.12-fold (*n* = 10) in amyotrophic lateral sclerosis (ALS), and to a 0.01-fold (*n* = 4) in spinal and bulbar muscular atrophy (SMA). (**b**) Changes in Shh signaling components shown in the SMA RNA-Seq data available from Rizzo et al. Changes in Shh and its key signaling components including GAS1, LRP2, SMO, PTCH1, SUFU, and Gli1, 2, and 3 are shown. (**c**) A summary of the Shh signaling pathway that leads to *Crabp1* activation. Shh signaling begins with LRP2 and GAS1-assisted release of PTCH1 inhibition of SMO. SMO facilitates the release of Gli1, 2, and 3 from inhibitor SUFU, allowing Gli activation to bind to their targets such as *Crabp1*. Open circles depict those components whose expression is reduced in SMA, including Shh, LRP2, PTCH1, SMO, SUFU, and Glis. One closed circle “●” depicts the component elevated in SMA because of a reduction in Shh. “#” marks *Crabp1* as the top gene out of 30 most significantly down-regulated genes in ALS motor neurons identified by Jiang et al. through GenePix Pro microarray analysis. For SMA RNA-seq data, ** *p*-value < 0.01, *** *p* -value < 0.001. Changes in PTCH1, SUFU, and Gli1–3 are not significant, but all exhibit the same trend of reduced expression. RNA-seq analysis was performed using CuffDiff2 Differential Analysis.

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
