# Peer review of "Sonic Hedgehog-Gli1 Signaling and Cellular Retinoic Acid Binding Protein 1 Gene Regulation in Motor Neuron Differentiation and Diseases"

_ijms, 2020, doi:10.3390/ijms21114125_

Round 1

Reviewer 1 Report

I have read the article “Sonic hedgehog-gli1 signaling and cellular retinoic acid binding protein1 gene regulation in motor neuron differentiation and diseases” by Lin et al.

The main background of the article is Shh and RA are required for neuronal differentiation, and that the activity of RA is modulated by the binding protein Crabp1, which is highly expressed in cholinergic neurons (motor neurons). Shh alone has no potential to transform ESC to motor neurons, yet alone is more effective that Shh and RA together in inducing Crabp1, an effect largely blocked by Gli inhibition. ChIP might suggest Gli binding to 2 distinct locations in the near promotor region of Crabp1. Two motor neuron diseases (ALS and SMA) are associated with abnormalities in Shh signaling.

The ChIP results are interesting, but I am not sufficiently versed in the technique, nor is there enough information provided, to ascertain whether the immunoprecipitation can distinguish binding to such nearby regions in the promoter when the output is determined by PCR rather than next gen sequencing. Perhaps this could be addressed. (That is, the beta hairpin of Gli may interact with the minimal Sp1 sites as well as its own ‘binding site’, or either alone. Can the technique differentiate these possibilities?)

My main concerns with the article are that the conclusions are inferential- associations without necessarily establishing causality.  Should the authors wish, they could consider the following:

  1. The necessity of RA, yet the induction of Crabp1 which might provide a form of feedback inhibition, suggest that relative levels of RA and Shh are critical in establishing an effect on differentiation. It would be interesting to know what Figure 3 would look like with different levels of RA (and/or ratios of RA to Shh).

  1. Similarly, the maximal effect on Crabp1 is seen with Shh alone (3c), yet the effect of Gli inhibition (3e) is presented only for the combined effect of Shh and RA. What is the effect of Gli inhibition on cells not exposed to RA (and hence not differentiated as per 3b), yet with highly induced Crabp1 (as per 3c)?

  1. I am unclear about the model indirectly proposed in 4d. Shh through Gli induces Crabp1. Does this increase nuclear concentrations of RA allowing RA to act at RA nuclear sites, or is Crabp1 a major sequestering protein reducing free levels of RA (and thereby decreasing nuclear activity)? What provides the temporal specificity for differentiation in the model?

  1. Does Crabp2 have similar roles?

  1. The concern with the literature survey (‘data mining’) is specificity. It may well be true that defects in Shh signaling lie near the heart of motor neuron diseases, but there is such a major difficulty with nucleocytoplasmic transport that one is concerned about specificity. There will be certainly widespread abnormalities in other pathways as well.

Author Response

IJMS803125 Critique/Response – Lin et al

Reviewer 1

I have read the article “Sonic hedgehog-gli1 signaling and cellular retinoic acid binding protein1 gene regulation in motor neuron differentiation and diseases” by Lin et al.

The main background of the article is Shh and RA are required for neuronal differentiation, and that the activity of RA is modulated by the binding protein Crabp1, which is highly expressed in cholinergic neurons (motor neurons). Shh alone has no potential to transform ESC to motor neurons, yet alone is more effective that Shh and RA together in inducing Crabp1, an effect largely blocked by Gli inhibition. ChIP might suggest Gli binding to 2 distinct locations in the near promotor region of Crabp1. Two motor neuron diseases (ALS and SMA) are associated with abnormalities in Shh signaling.

The ChIP results are interesting, but I am not sufficiently versed in the technique, nor is there enough information provided, to ascertain whether the immunoprecipitation can distinguish binding to such nearby regions in the promoter when the output is determined by PCR rather than next gen sequencing. Perhaps this could be addressed. (That is, the beta hairpin of Gli may interact with the minimal Sp1 sites as well as its own ‘binding site’, or either alone. Can the technique differentiate these possibilities?)

---- Response:

ChIP-qPCR has been widely used to determine specific protein-binding region on the chromatin target in vivo. In fact, this procedure was first developed and established in our lab, as published in a study published in Mol Cell [1] as well as numerous later studies of our own and others [2-7] including applications in genome-wide survey studies mentioned by this reviewer known as ChIP-seq [8]. ChIP-qPCR is most accurate and precise when a specific target chromatin is predicted (in our case the target is a particular chromatin region, the Crabp1 promoter). ChIP-seq, however, is a genome wide survey for potential binding chromatin targets. We believe our question should be best addressed using targeted ChIP-qPCR in our study, because we are addressing specific Gli1 and SP1 binding sites in the defined chromatin region, the Crabp1 promoter. These technical references have been added.

My main concerns with the article are that the conclusions are inferential- associations without necessarily establishing causality.  

--- Response:

Yes, the goal of this paper is to present the first evidence for a motor neuron-activating effect associated with Shh/Gli signal input, in combination with RA, which together regulates a proper level of Crabp1 expression; importantly, this association, Shh/Gli signaling with Crabp1 in neurological diseases, is supported by available human data. We like to point out that in the Abstract, we did summarize our conclusion, twice, with descriptors like “correlation” (line 20) and “correlated” (line 29), as well as numerous statements throughout the text. We further emphasized this point in our modified Discussion (in the first paragraph, lines 325-328). We also briefly discussed the need to study the functional role of Crabp1 in the context of motor neuron activation in the future (the last paragraph in Discussion, lines 382-384).

What the reviewer asked in the following sub-questions are all legitimate, but will require a very comprehensive, systematic series of genetic, molecular, cell biology and physiology experiments that are really not practically feasible. We hope the reviewer understands that we appreciate the points, and we are ultimately aiming to address these important questions in subsequent studies.

Should the authors wish, they could consider the following.

The necessity of RA, yet the induction of Crabp1 which might provide a form of feedback inhibition, suggest that relative levels of RA and Shh are critical in establishing an effect on differentiation. It would be interesting to know what Figure 3 would look like with different levels of RA (and/or ratios of RA to Shh)

Similarly, the maximal effect on Crabp1 is seen with Shh alone (3c), yet the effect of Gli inhibition (3e) is presented only for the combined effect of Shh and RA. What is the effect of Gli inhibition on cells not exposed to RA (and hence not differentiated as per 3b), yet with highly induced Crabp1 (as per 3c)?

--- Response: please see response provided above.

I am unclear about the model indirectly proposed in 4d. Shh through Gli induces Crabp1. Does this increase nuclear concentrations of RA allowing RA to act at RA nuclear sites, or is Crabp1 a major sequestering protein reducing free levels of RA (and thereby decreasing nuclear activity)? What provides the temporal specificity for differentiation in the model?

--- Response:

The model provided in Fig. 4d in fact has nothing to do with Crabp1 function. This figure is to describe molecular events that may occur on Crabp1 promoter where Gli (Shh down stream signaling) executes its action to regulate Crabp1 gene by working with Sp1 to access the PIC (pre-initiation complex for transcription), a key step in activating Crabp1 gene expression. Legend of Fig. 4d has been made clear that, this figure depicts a “molecular model….of Crabp1 gene” (lines 300-302).

As to how Crabp1 functions, either via its presumed, canonical, RA-sequestering activity or its recently established non-canonical cytoplasmic activity, this awaits further study. The current study merely describes the association of Shh/Gli (and RA) via regulating Crabp1 gene with motor neuron differentiation, which is supported by available human data. This has been added to the end of Discussion (lines 382-384).

Does Crabp2 have similar roles?

--- Response:

Although CRABP2 shares some properties with CRABP1 (such as sequence and RA binding), the promoter of Crabp2 gene does not have a Gli binding site, nor does it have multiple Sp1 sites. Crabp2 is not related to Shh/Gli signaling pathway in the context of motor neuron and Crabp2 is a totally different gene, irrelevant to this current study.

The concern with the literature survey (‘data mining’) is specificity. It may well be true that defects in Shh signaling lie near the heart of motor neuron diseases, but there is such a major difficulty with nucleocytoplasmic transport that one is concerned about specificity. There will be certainly widespread abnormalities in other pathways as well.

--- Response:

We absolutely agree with this reviewer, motor neuron diseases are complex, and cannot be just specifically related to one protein, one signaling pathway or just the expression of a gene. This current study reports a new finding of the association of Crabp1 elevation by Shh/Gli with motor neuron differentiation, which is supported by, and correlated with, available data of human neurological diseases.

References

  1. Park, S. W.; Li, G.; Lin, Y. P.; Barrero, M. J.; Ge, K.; Roeder, R. G.; Wei, L. N., Thyroid hormone-induced juxtaposition of regulatory elements/factors and chromatin remodeling of Crabp1 dependent on MED1/TRAP220. Molecular cell 2005, 19, (5), 643-53.
  2. Lee, B. H.; Stallcup, M. R., Glucocorticoid receptor binding to chromatin is selectively controlled by the coregulator Hic-5 and chromatin remodeling enzymes. The Journal of biological chemistry 2017, 292, (22), 9320-9334.
  3. Conrad, R. J.; Fozouni, P.; Thomas, S.; Sy, H.; Zhang, Q.; Zhou, M. M.; Ott, M., The Short Isoform of BRD4 Promotes HIV-1 Latency by Engaging Repressive SWI/SNF Chromatin-Remodeling Complexes. Molecular cell 2017, 67, (6), 1001-1012 e6.
  4. Du, C.; Jin, Y. Q.; Qi, J. J.; Ji, Z. X.; Li, S. Y.; An, G. S.; Jia, H. T.; Ni, J. H., Effects of myogenin on expression of late muscle genes through MyoD-dependent chromatin remodeling ability of myogenin. Molecules and cells 2012, 34, (2), 133-42.
  5. Burd, C. J.; Ward, J. M.; Crusselle-Davis, V. J.; Kissling, G. E.; Phadke, D.; Shah, R. R.; Archer, T. K., Analysis of chromatin dynamics during glucocorticoid receptor activation. Molecular and cellular biology 2012, 32, (10), 1805-17.
  6. Persaud, S. D.; Huang, W. H.; Park, S. W.; Wei, L. N., Gene repressive activity of RIP140 through direct interaction with CDK8. Mol Endocrinol 2011, 25, (10), 1689-98.
  7. Park, S. W.; Huang, W. H.; Persaud, S. D.; Wei, L. N., RIP140 in thyroid hormone-repression and chromatin remodeling of Crabp1 gene during adipocyte differentiation. Nucleic acids research 2009, 37, (21), 7085-94.
  8. Nakato, R.; Sakata, T., Methods for ChIP-seq analysis: A practical workflow and advanced applications. Methods 2020.
  9. Liyang, G.; Abdullah, S.; Rosli, R.; Nordin, N., Neural Commitment of Embryonic Stem Cells through the Formation of Embryoid Bodies (EBs). Malays J Med Sci 2014, 21, (5), 8-16.
  10. Guan, K.; Chang, H.; Rolletschek, A.; Wobus, A. M., Embryonic stem cell-derived neurogenesis. Retinoic acid induction and lineage selection of neuronal cells. Cell and tissue research 2001, 305, (2), 171-6.
  11. Wichterle, H.; Lieberam, I.; Porter, J. A.; Jessell, T. M., Directed differentiation of embryonic stem cells into motor neurons. Cell 2002, 110, (3), 385-97.

Reviewer 2 Report

In this manuscript the authors study the regulation of CRABP1 in motor neurons. They first show that this gene is expressed in motor neurons in the adult spinal cord and downregulated in motor neurons transfected with mutant SOD1 or androgen receptor variants that cause amyotrophic lateral sclerosis or spinal muscular atrophy and in different types of sequencing data from patient-derived motor neurons. To better understand the molecular signals that drive CRABP1 expression in motor neurons, they generate motor neurons in-vitro and show that Shh, a ventralising signal in the developing spinal cord, promotes the expression of CRABP1 in their in-vitro cultures. They validate this model further by demonstrating direct binding of Gli1 to a predicted Gli-binding site in the CRABP1 in close proximity to the transcription initiation site.

Broad comments:

There are some problems with this manuscript in my opinion. First, down-regulation of CRABP1 in SMA and ALS motor neurons has been demonstrated before (in studies which the authors cite appropriately), so the results presented in Fig. 2 and Fig. 5A are more a confirmation of previous results than truly novel. Furthermore, the authors propose a model in which down-regulated Shh signalling is responsible for the reduced expression of CRABP1 in diseased motor neurons, but do not provide any experimental evidence for this. Does, for example, activation of the Shh pathway in diseased motor neurons rescue the down-regulation of CRABP1 presented in Figure 2? The finding that Gli1 binds to the promoter proximal region of the CRABP1 gene is interesting though.

Specific comments:

Fig. 1: The authors should better quantify this. What is the percentage of motor neurons that express CRABP1? All of them or only a subsection, and if the latter then which one? Also, is CRABP1 expressed in other types of neurons in the spinal cord, e.g. excitatory or inhibitory interneurons?

Figure 2: What precisely was done here? If I am not mistaken this is not described in the Materials and Methods section.

Fig. 3: There are several problems here. In the text, the authors say: “To understand how Crabp1 gene is highly and specifically activated in motor neurons, weemployed an ESC-motor neuron differentiation model.”, however, in the figure the authors show the strongest induction of CRABP1 in the Shh only group, in which the motor neuron markers Hb9 and Chat are not induced. Do the authors know the proportion of motor neurons they generate under each of these conditions and what kind of other cell types they may generate in the Shh only condition? Similarly, for the GANT61 treatments the main effect seems to be that motor neuron differentiation is suppressed and not CRABP1 expression in motor neurons. The authors should disentangle these two processes, as showing conclusively that GANT61 treatment down-regulates CRABP1 specifically in motor neurons would strongly support their model that Shh signaling in motor neurons is required for CRABP1 expression.

Fig. 5: The authors should add error bars or similar to these plots, or are these single measures?

Author Response

IJMS803125 Critique/Response – Lin et al

Reviewer 2

In this manuscript the authors study the regulation of CRABP1 in motor neurons. They first show that this gene is expressed in motor neurons in the adult spinal cord and downregulated in motor neurons transfected with mutant SOD1 or androgen receptor variants that cause amyotrophic lateral sclerosis or spinal muscular atrophy and in different types of sequencing data from patient-derived motor neurons. To better understand the molecular signals that drive CRABP1 expression in motor neurons, they generate motor neurons in-vitro and show that Shh, a ventralising signal in the developing spinal cord, promotes the expression of CRABP1 in their in-vitro cultures. They validate this model further by demonstrating direct binding of Gli1 to a predicted Gli-binding site in the CRABP1 in close proximity to the transcription initiation site.

Broad comments:

There are some problems with this manuscript in my opinion. First, down-regulation of CRABP1 in SMA and ALS motor neurons has been demonstrated before (in studies which the authors cite appropriately), so the results presented in Fig. 2 and Fig. 5A are more a confirmation of previous results than truly novel. Furthermore, the authors propose a model in which down-regulated Shh signalling is responsible for the reduced expression of CRABP1 in diseased motor neurons, but do not provide any experimental evidence for this. Does, for example, activation of the Shh pathway in diseased motor neurons rescue the down-regulation of CRABP1 presented in Figure 2? The finding that Gli1 binds to the promoter proximal region of the CRABP1 gene is interesting though.

--- Response: Thank you for these meaningful suggestions.

We like to emphasize that the goal of this study is NOT to conclude a causative role for down regulated Shh (therefore reduction in Crabp1) in motor neuron diseases. The goal of this report is to provide the first evidence for Shh regulation of Crabp1, which can be associated with motor neuron differentiation is supported by available human motor neuron disease data.

Comprehensive, systemic studies are absolutely needed in the future to unambiguously prove this pathway, which, we humbly believe, are practically beyond the scope of this paper.

Specific comments:

Fig. 1: The authors should better quantify this. What is the percentage of motor neurons that express CRABP1? All of them or only a subsection, and if the latter then which one? Also, is CRABP1 expressed in other types of neurons in the spinal cord, e.g. excitatory or inhibitory interneurons?

--- Response:

In our preliminary study, we found that all Crabp1+ cells in spinal sections are NeuN+, indicating that all Crabp1-expressing cells in the spinal section are neurons. However, not all NeuN+ cells are Crabp1+ (white arrow), indicating that Crabp1 is expressed in specific subsets of neurons in the spinal cord. This data is now provided as Supplementary Fig. 1 (ventral horn in the upper half of the image, dorsal horn in the lower half of the image.

We further found that almost all motor neurons (ChAT+) in the lumbar spinal cord express Crabp1 (Crabp1+), therefore we focused on motor neurons in this study. In the main text, Fig. 1b shows a lumbar spinal cord section stained with Crabp1, a motor neuron marker ChAT, a non-neuron marker GFAP and a nuclear marker DAPI. It appears that ChAT+ cells (motor neurons) are almost 100% overlapping with Crabp1+ cells, but none of GFAP+ cells (non-neurons) overlap with Crabp1+ cells. As suggested, we scored ChAT+ overlapping with Crabp1+ cells. The quantification is shown as a bar graph in Supplementary Data Fig. 2.

These new data are now described in text under section 2.1 (lines 113-126).

Figure 2: What precisely was done here? If I am not mistaken this is not described in the Materials and Methods section.

--- Response:

We apologize for this mistake. In section 4.4, we have modified this section to provide detailed method of Figure 2 (4.4 Cell culture and the disease models, lines 416-429).

Fig. 3: There are several problems here. In the text, the authors say: “To understand how Crabp1 gene is highly and specifically activated in motor neurons, we employed an ESC-motor neuron differentiation model.”, however, in the figure the authors show the strongest induction of CRABP1 in the Shh only group, in which the motor neuron markers Hb9 and Chat are not induced. Do the authors know the proportion of motor neurons they generate under each of these conditions and what kind of other cell types they may generate in the Shh only condition?

--- Response:

In the will established ESC-neuronal differentiation model, RA is the key agent to induce EB differentiation into neurons [9, 10]. Shh further facilitates RA-exposed EB differentiation into motor neurons [11]. Shh alone is not able to induce motor neurons (as supported by the lack of ChAT and Hb9 expression), although it does activate Crabp1 expression (as shown in this figure). This type of culture system merely provides an in vitro tool to dissect the involved pathways/factors, as we wish to demonstrate Shh-activation of Crabp1 in this culture model. In fact, this result further confirms the widely known complexity of neuronal differentiation/specialization, and supports our proposal that motor neuron differentiation involves multiple players (at least Shh/Gli, RA and Crabp1, as revealed in this study) to engage specific pathways in completing the differentiation/activaton process. Text in section 2.3 (for Fig 2, lines 162-166) has been modified to improve the clarity of this ESC-motor neuron differentiation system.

Similarly, for the GANT61 treatments the main effect seems to be that motor neuron differentiation is suppressed and not CRABP1 expression in motor neurons. The authors should disentangle these two processes, as showing conclusively that GANT61 treatment down-regulates CRABP1 specifically in motor neurons would strongly support their model that Shh signaling in motor neurons is required for CRABP1 expression.

--- Response:

We agree with this reviewer. Abundant literature has shown complicated Shh effects, which definitely are not limited to motor neurons. Another complication came from the fact that, Crabp1 can also be regulated by multiple pathways including thyroid hormones, RA, epigenetic (DNA methylation), in addition to the new finding of Shh pathway reported in this current study. In an in vitro differentiation culture system, multiple cell types are expected. In Fig 3b and c, we intend to demonstrate Shh as one key factor in up-regulating Crabp1 gene in this motor neuron differentiation process, which can be inhibited by a specific inhibitor of Shh down stream signal mediator Gli, GANT61. This provides a logical connection for the Shh-Gli signaling pathway that regulates Crabp1 expression. Therefore, this section is then followed by the next experiment showing Gli1 binding to Crabp1 promoter, supporting the effect of Shh.

Fig. 5: The authors should add error bars or similar to these plots, or are these single measures?

--- Response:

We agree. However, in Figure 5A, standard deviation and standard error were not made available in the human dataset to create error bars. In the ALS study, Jiang et al calculated the fold change from the mean values of motor neuron from 5 ALS patients divided by 5 healthy individuals, for a total n=10. Microarray data was analyzed by GenePix Pro to identify significantly reduced genes in ALS. Reduced Crabp1 expression was validated through quantitative RT-PCR in both ALS and SMA studies.

In Figure 5B, standard deviation and standard error were not made available to create error bars either. In the SMA study, Rizzo et al generated the FKPM values from motor neurons derived from 2 individual SMA patients and 2 healthy controls for a total n=4. The RNA-seq data were analyzed with the Cuffdiff2 algorithm for significant differences in gene expression in healthy versus SMA motor neurons.

The Methods section 4.8 has been updated to clarify the number of measures taken in the ALS study and SMA study (lines 488-490; 496-497). Result section 2.5 (lines 245-246) was also modified to emphasize that reduced Crabp1 expression seen in the ALS microarray study and RNA-seq SMA experiments were further validated by quantitative RT-PCR.

References

  1. Park, S. W.; Li, G.; Lin, Y. P.; Barrero, M. J.; Ge, K.; Roeder, R. G.; Wei, L. N., Thyroid hormone-induced juxtaposition of regulatory elements/factors and chromatin remodeling of Crabp1 dependent on MED1/TRAP220. Molecular cell 2005, 19, (5), 643-53.
  2. Lee, B. H.; Stallcup, M. R., Glucocorticoid receptor binding to chromatin is selectively controlled by the coregulator Hic-5 and chromatin remodeling enzymes. The Journal of biological chemistry 2017, 292, (22), 9320-9334.
  3. Conrad, R. J.; Fozouni, P.; Thomas, S.; Sy, H.; Zhang, Q.; Zhou, M. M.; Ott, M., The Short Isoform of BRD4 Promotes HIV-1 Latency by Engaging Repressive SWI/SNF Chromatin-Remodeling Complexes. Molecular cell 2017, 67, (6), 1001-1012 e6.
  4. Du, C.; Jin, Y. Q.; Qi, J. J.; Ji, Z. X.; Li, S. Y.; An, G. S.; Jia, H. T.; Ni, J. H., Effects of myogenin on expression of late muscle genes through MyoD-dependent chromatin remodeling ability of myogenin. Molecules and cells 2012, 34, (2), 133-42.
  5. Burd, C. J.; Ward, J. M.; Crusselle-Davis, V. J.; Kissling, G. E.; Phadke, D.; Shah, R. R.; Archer, T. K., Analysis of chromatin dynamics during glucocorticoid receptor activation. Molecular and cellular biology 2012, 32, (10), 1805-17.
  6. Persaud, S. D.; Huang, W. H.; Park, S. W.; Wei, L. N., Gene repressive activity of RIP140 through direct interaction with CDK8. Mol Endocrinol 2011, 25, (10), 1689-98.
  7. Park, S. W.; Huang, W. H.; Persaud, S. D.; Wei, L. N., RIP140 in thyroid hormone-repression and chromatin remodeling of Crabp1 gene during adipocyte differentiation. Nucleic acids research 2009, 37, (21), 7085-94.
  8. Nakato, R.; Sakata, T., Methods for ChIP-seq analysis: A practical workflow and advanced applications. Methods 2020.
  9. Liyang, G.; Abdullah, S.; Rosli, R.; Nordin, N., Neural Commitment of Embryonic Stem Cells through the Formation of Embryoid Bodies (EBs). Malays J Med Sci 2014, 21, (5), 8-16.
  10. Guan, K.; Chang, H.; Rolletschek, A.; Wobus, A. M., Embryonic stem cell-derived neurogenesis. Retinoic acid induction and lineage selection of neuronal cells. Cell and tissue research 2001, 305, (2), 171-6.
  11. Wichterle, H.; Lieberam, I.; Porter, J. A.; Jessell, T. M., Directed differentiation of embryonic stem cells into motor neurons. Cell 2002, 110, (3), 385-97.

Round 2

Reviewer 2 Report

The authors have addressed several of my criticisms, e.g. the quantification of Crabp1 expression in motor neurons and the omissions in the Material and Methods section, so I think the manuscript can be published in its current form.

I still think that a conclusive demonstration that inhibition of Shh signalling in motor neurons leads to loss of Crabp1 expression would have given this study much broader significance . At the moment it rather looks like that the reduction they see in their GANT61 treatments (Fig. 3d,e) stems from a change in cellular identity. However, as a first indication that Shh signalling is involved in the regulation of Crabp1 expression, the results presented in this manuscript are sufficient.